# Bimekizumab in the Treatment of Axial Spondyloarthritis and Psoriatic Arthritis: A New Kid on the Block

**DOI:** 10.3390/ijms26052315

**Published:** 2025-03-05

**Authors:** Julie Sarrand, Laurie Baglione, Charlotte Bouvy, Muhammad Soyfoo

**Affiliations:** Department of Rheumatology, Hôpital Universitaire de Bruxelles, Université Libre de Bruxelles, 1070 Bruxelles, Belgium; julie.sarrand@ulb.be (J.S.); laurie.baglione@hubruxelles.be (L.B.); charlotte.bouvy@ulb.be (C.B.)

**Keywords:** interleukin-17 (IL-17), bimekizumab, psoriasis (PsO), psoriatic arthritis (PsA), axial spondyloarthritis (axSpA), cytokines, IL-17A, IL-17F, inflammation

## Abstract

The interleukin (IL)-17 family encompasses six structurally related pro-inflammatory cystine knot proteins, designated as IL-17A to IL-17F. Over the last decades, evidence has pointed to its role as a critical player in the development of inflammatory diseases such as psoriasis (PsO), axial spondyloarthritis (axSpA), and psoriatic arthritis (PsA). More specifically, IL-17A and IL-17F are overexpressed in the skin and synovial tissues of patients with these diseases, and recent studies suggest their involvement in promoting inflammation and tissue damage in axSpA and PsA. Bimekizumab is a monoclonal antibody targeting both IL-17A and IL-17F, playing an important role in the treatment of these diseases. This review details the implications of bimekizumab in the therapeutic armamentarium of axSpA and PsA.

## 1. Introduction

Spondyloarthropathies (SpAs) is an umbrella term for inflammatory diseases, including axial spondyloarthritis (axSpA) and psoriatic arthritis (PsA). These diseases are characterized by elevated inflammatory markers, the absence of autoantibodies (seronegative diseases), and a strong association with the human leukocyte antigen (HLA)-B27 genotype. They are often accompanied by extra-articular manifestations such as uveitis, psoriasis, and inflammatory bowel disease (IBD). However, the frequency and clinical relevance of these manifestations vary, reflecting the heterogeneity within the spectrum of spondyloarthritis [1].

AxSpA is further classified into radiographic axSpA (r-axSpA), also known as ankylosing spondylitis (AS), and non-radiographic axSpA (nr-axSpA) based on the presence or absence of sacroiliitis on standard radiographs [2]. Clinically, both r-axSpA and nr-axSpA are defined by chronic pain and stiffness in the axial skeleton, which can progress to ankylosis. Additional manifestations include enthesitis, observed in 40–60% of patients, acute anterior uveitis in 30–50%, and colitis, which may be symptomatic in 15% and microscopic in up to 60% [3].

PsA is primarily characterized by peripheral arthritis, enthesitis, and dactylitis, but it can also involve the axial skeleton and be associated with uveitis and IBD. Psoriasis occurs in approximately 25% of patients with PsA, usually preceding the onset of arthritis. However, in 20% of cases, arthritis and psoriasis appear simultaneously, and arthritis may precede the diagnosis or recognition of psoriasis in up to 15% of patients [4].

The primary objective of SpA treatments is to restrain inflammation and hamper structural damage, thereby reducing morbidity and enhancing patients’ quality of life. Given the heterogeneity in disease manifestations, the selection of treatment is tailored to individual patient characteristics, potential contraindications, and expected efficacy based on the specific type of involvement [5,6].

For patients with axSpA, non-steroidal anti-inflammatory drugs (NSAIDs) remain the anchor drug of first-line therapy. However, their therapeutic efficacy may be insufficient, and associated adverse effects may limit their long-term use in numerous cases [7]. Among conventional disease-modifying antirheumatic drugs (csDMARDs), including methotrexate, sulfasalazine, and leflunomide, only sulfasalazine has demonstrated efficacy in addressing peripheral manifestations such as enthesitis [8]. For patients who exhibit inadequate responses or intolerance to NSAIDs, biological DMARDs (bDMARDs), and targeted synthetic DMARDs (tsDMARDs)—including inhibitors of tumor necrosis factor-alpha (TNF-α), IL-17, and Janus kinase (JAK)—have all been validated for this indication [5].

In PsA, the first-line therapy primarily involves csDMARDs. In cases of therapeutic failure, bDMARDs, such as anti-TNF-α agents, IL-17 inhibitors, and cytotoxic T-lymphocyte-associated protein 4 (CTLA-4) inhibitors, as well as tsDMARDs, including phosphodiesterase-4 (PDE4) and JAK inhibitors, are all effective options [9]. Despite this wide range of therapeutic options, a significant number of patients fail to achieve remission, supporting the necessity for novel therapeutic options [10,11].

To this day, the precise pathophysiology of SpA remains not yet fully understood; it is likely the result of a complex interplay between genetic predisposition, environmental factors, the microbiome, and/or biomechanical stress. Notably, the gastrointestinal system has been implicated in the inflammatory burden of spondyloarthropathies, as a significant proportion of patients present with concomitant IBD. Even in the absence of overt IBD, subclinical intestinal inflammation is frequently observed [12]. Mechanical stress is another risk factor, particularly when interacting with genetic predisposition, which can trigger disease onset and drive joint inflammation as well as aberrant bone repair, ultimately leading to pathological new bone formation at entheses [13].

Compelling evidence underlines the critical role of the IL-23/IL-17 axis in the pathogenesis of SpA. The binding of IL-23 to its receptor induces the upregulation of the transcription factor RORγt, leading to the production of various chemokines and cytokines, including TNF-α, IL-21, IL-26, and, most particularly, IL-17. This mechanistic insight explains the potent efficacy of several classes of bDMARDs in alleviating SpA disease activity. Additionally, the dysregulation of IL-17A and IL-17F has been shown to drive inflammation and excessive bone formation. Furthermore, the strongest genetic association with spondyloarthropathies is HLA-B27, which, through multiple mechanisms, fosters the production of IL-23 and IL-17 [14]. Finally, elevated levels of IL-17-producing cells are disproportionately prevalent in the blood, entheses, and synovial fluid of patients with SpA [15,16,17].

Recent studies show that targeting both IL-17A and IL-17F may be more effective than targeting IL-17A alone in treating these diseases [18]. Bimekizumab, a humanized monoclonal IgG1 antibody that neutralizes both IL-17A and IL-17F, has shown substantial efficacy in the treatment of psoriasis (PSO), psoriatic arthritis (PsA), and AxSpA. Bimekizumab’s efficacy was successively demonstrated in treating active PsA in the phase 3 BE COMPLETE study [19] and in the phase 3 BE Mobile 1 and 2 study, which demonstrated the potential of the dual inhibition of IL-17A and IL-17F to improve clinical outcomes in patients with active axSpA (r and nr-AxSpA) [20]. Bimekizumab (Bimzelx^®^, UCB, Brussels, Belgium) received the European Medicines Agency (EMA)’s approval on 20 August 2021, for treating moderate to severe plaque psoriasis in adults who need systemic therapy. Its indications were later expanded to include PsA, axSpA, and active moderate to severe hidradenitis suppurativa (HS). For PsA, it can be used alone or with methotrexate in adults with inadequate response or intolerance to DMARDs. In axSpA, it is approved for active disease in adults with signs of inflammation who have not responded to NSAIDs, including both non-radiographic and ankylosing spondylitis. Bimzelx is also approved for HS in adults with an inadequate response to conventional therapy.

The pathogenesis of both axSpA and PsA is intricately linked to the IL-23–IL-17 pathway, with IL-17A and IL-17F playing central roles in driving inflammation, joint damage, and abnormal bone formation or erosion. While axSpA is characterized by new bone formation and inflammation at the entheses, PsA features a combination of joint inflammation, bone erosion, and skin involvement. Targeting both IL-17A and IL-17F through dual blockade offers promising therapeutic strategies for controlling inflammation and preventing disease progression in both conditions. However, the differences in clinical manifestations underscore the need for tailored treatments to address the unique challenges presented by each disease.

This narrative review aims to detail the implications of bimekizumab in the therapeutic armamentarium of axSpA and PsA. For that purpose, we conducted a comprehensive literature search on PubMed for studies published between 2007 and 2025, using keywords such as “bimekizumab”, “IL-17”, “IL-17A”, “IL-17F”, “psoriatic arthritis”, and “ankylosing spondylitis”. Studies were selected based on their relevance to the therapeutic effects of bimekizumab in IL-17 inhibition. Clinical trials, observational studies, and systematic reviews were included if they reported on the safety, efficacy, and mechanisms of bimekizumab in these inflammatory diseases. Only English-language studies were considered. The data were synthesized narratively to summarize the current evidence and highlight areas for future research.

## 2. The IL-17 Pathway and Its Role in the Pathophysiology of Spondyloarthropathies

### 2.1. IL-17 Cytokine Family and Related Pathways

IL-17A, also referred to as IL-17, was the first member of the IL-17 cytokine family to be identified in the mid-1990s [21,22]. This groundbreaking discovery facilitated the identification of five additional isoforms that collectively constitute the IL-17 family: IL-17B, IL-17C, IL-17D, IL-17E (commonly known as IL-25), and IL-17F. Structurally, IL-17 cytokines are distinguished by their unique cysteine-knot fold, a hallmark feature that differentiates them from other cytokine subclasses [23].

The IL-17 receptor (IL-17R) family comprises five subunits: IL-17RA, IL-17RB, IL-17RC, IL-17RD, and IL-17RE. These receptors are unified by the presence of a conserved cytoplasmic SEFIR (SEF/IL-17 receptor) motif, which shares a distant evolutionary relationship with the Toll-IL-1 receptor (TIR) domain [24].

Among the IL-17 family, IL-17A and IL-17F are the most extensively characterized, sharing the highest sequence homology (55%). These cytokines function as homo- or heterodimers that engage a heterodimeric receptor complex composed of IL-17RA and IL-17RC [25].

IL-17A homodimers elicit the most robust responses, followed by IL-17A/F heterodimers and IL-17F homodimers [26,27]. Upon receptor activation, canonical signaling cascades, including Nuclear factor kappa-light-chain-enhancer of activated B cells (NF-κB), mitogen-activated protein (MAP) kinases, and CCAAT/enhancer-binding protein-beta (C/EBP-β), are triggered [28]. This results in the transcriptional activation of pro-inflammatory mediators, such as IL-6, IL-8, and TNF-α [29,30]. Beyond its pro-inflammatory functions, the activated IL-17 receptor also interacts with other cell surface receptors, including the epidermal growth factor receptor (EGFR) and fibroblast growth factor receptor (FGFR), thereby influencing biological processes such as cell proliferation and tissue remodeling [31].

The primary source of IL-17A and IL-17F is the T helper 17 (Th17) subset of CD4+ T cells, whose differentiation is regulated by a network of cytokines, including IL-23, tumor growth factor-β (TGF-β), IL-6, IL-1β, and IL-21 [32]. Notably, IL-23 is essential for stabilizing and maintaining the effector functions of Th17 cells [33]. Conversely, cytokines, such as interferon (IFN)-γ, IL-12, and IL-4, inhibit Th17 differentiation, redirecting naïve T cells toward Th1 or Th2 lineages. In addition to Th17 cells, IL-17A and IL-17F are secreted by various innate immune cell subsets, including γ/δ T cells, innate lymphoid cells 3 (ILC3), neutrophils, and mast cells [34]. Of these, γ/δ T cells are particularly abundant in anatomical niches prone to inflammatory pathology, such as the ciliary body, colon, and entheses, which are frequently implicated in spondyloarthropathies [35].

Functionally, IL-17 cytokines exhibit notable pleiotropy, with roles extending beyond their classical pro-inflammatory effects. These cytokines are integral to both physiological and pathological processes [36,37].

While IL-17A and IL-17F are critical for host defense against bacterial and fungal pathogens [37,38,39], they also contribute to mucosal integrity, allergen responses, and the regulation of lymphocyte activity [36,40,41]. IL-17 cytokines recruit neutrophils and macrophages to sites of inflammation, amplifying immune responses and perpetuating tissue damage in chronic inflammatory conditions. Dysregulated IL-17 signaling has been implicated in the pathogenesis of autoimmune diseases, leading to excessive inflammation, tissue remodeling, and fibrosis [42,43,44].

### 2.2. Role of IL-17 in axSpA Physiopathology

Over the past decade, substantial evidence has underscored the critical role of the IL-23/IL-17 axis in the pathophysiology of axSpA. Elevated levels of IL-17A and IL-17F in the serum and tissues of axSpA patients contribute significantly to disease progression [45]. IL-17A induces the expression of matrix metalloproteinases (MMPs), such as MMP1, MMP9, and MMP13, which degrade the extracellular matrix and cause tissue damage [46]. Notably, elevated MMP-3 levels have emerged as a biomarker of disease activity, correlating with the clinical severity and structural damage observed in imaging studies [47]. Additionally, IL-17A stimulates receptor activator of nuclear factor kappa-Β ligand, leading to osteoclast activation, bone resorption, and bone loss. IL-17A also promotes angiogenesis, facilitating inflammatory cell infiltration and amplifying local inflammation [48]. While IL-17F is less potent than IL-17A, it complements IL-17A’s effects, particularly in later disease stages [49].

HLA-B27, the strongest genetic association in axSpA, may contribute to disease through peptide presentation, homodimerization, or protein misfolding, triggering immune responses [50]. Other major histocompatibility complex (MHC) class I loci, such as HLA-A02 and HLA-B07, are associated with axSpA [51]. Additionally, genetic variants in runt-related transcription factor 3 (RUNX3) and endoplasmic reticulum aminopeptidase 1/2 (ERAP1/2), which regulate T cell activity and MHC class I peptide processing, are linked to axSpA [52,53]. Variants in IL-23R, IL-23A, IL12-B, tyrosine kinase 2 (TYK2), TRAF3 Interacting Protein 2 (TRAF3IP2), signal transducer and activator of transcription 3 (STAT3), and TNF alpha-induced protein 3 (TNFAIP3), further implicate the IL-23–IL-17 axis and NF-κB signaling in disease pathogenesis [54,55] (Figure 1).

In axSpA, IL-17A and TNF play central roles in ectopic bone formation, leading to syndesmophytes and spinal ankylosis, hallmark features of the disease. IL-17F supports inflammation in later stages but is secondary to IL-17A in driving pathology. Elevated IL-17-producing T cells, including CD4+ (Th17) and CD8+ (Tc17) subsets, are found in axSpA and correlate with disease markers such as C-reactive protein (CRP) levels [56].

In axSpA, serum IL-17 levels correlate with disease activity as measured by the Bath Ankylosing Spondylitis Disease Activity Index (BASDAI) score [57]. IL-17 levels are significantly higher in the synovial fluid compared to serum in patients with undifferentiated SpA [58]. Increased IL-23A transcription in intestinal biopsies of axSpA patients suggests a potential gut-derived IL-23 source, even though IL-17 messenger RNA (mRNA) levels are not similarly elevated. IL-17A and TNF jointly promote mesenchymal stem cell differentiation into osteoblasts, driving ectopic bone formation in SpA [12].

IL-17+CD4+ and IL-17+CD8+ T cells are enriched in the synovial fluid and peripheral blood of axSpA patients compared to controls, correlating with disease activity [59,60]. This suggests that IL-17+CD8+ T cells are linked to HLA class I-associated SpA and severe disease phenotypes [61,62].

Mucosal-associated invariant T (MAIT) cells, innate-like T cells with unique T cell receptor features, can produce IL-17, TNF, IFNγ, and granzyme B upon activation. Higher frequencies of IL-17+ MAIT cells are observed in the peripheral blood of axSpA patients compared to healthy controls, further highlighting the contribution of the IL-23/IL-17 axis to axSpA [63].

### 2.3. IL-17’s Role in PsA

PsA is driven by complex interactions between cytokines, immune cells, and genetic factors, with IL-17A and IL-17F playing pivotal roles in its pathogenesis. Together, IL-17A and IL-17F synergize to amplify synovial inflammation and joint damage [64]. Additionally, TNF and IL-17A are central to synovial inflammation and bone erosion in PsA, contributing to joint destruction rather than the new bone formation commonly seen in axSpA [49].

PsA shares genetic predispositions with related conditions such as psoriasis and IBD, involving genes like IL23A, IL12B, and IL23R. The strongest genetic association in PsA and axSpA overall is with HLA-B27, along with other MHC class I loci such as HLA-B39, HLA-Cw6, and HLA-B38 [55,65]. These genetic factors contribute to dysregulated IL-23 and IL-17 pathways, which drive inflammation in both the skin and joints.

Immune cells play a central role in PsA pathogenesis. TH17 cells, enriched in psoriatic skin lesions and inflamed joints, produce IL-17A, IL-17F, and IL-22, driving synovial and skin inflammation [66]. Elevated levels of IL-17+CD4+ T cells are found in the peripheral blood and synovial fluid of PsA and axSpA patients, correlating with markers of inflammation such as CRP and ESR [28]. Tissue-resident memory (TRM) T cells, localized in tissues rather than circulating in the blood, are also involved in chronic inflammation. Found in psoriatic lesions and inflamed joints, TRM T cells produce IL-17A, IFN-γtr, and IL-22. These cells, marked by CD69 and/or CD103, are regulated by the RUNX3 transcription factor and persist in tissues to sustain inflammation [67].

IL-17+CD8+ T cells are another critical component, enriched in the synovial fluid of PsA and axSpA patients, where they correlate with disease activity markers such as CRP and imaging results [62]. In psoriasis, these cells are prominent in epidermal lesions, directly contributing to skin inflammation and disease severity. γδ T cells, which have hybrid innate–adaptive functions, are rapidly activated to produce IL-17A, IL-17F, and IL-22 [68]. These cells are enriched in PsA synovial fluid and play key roles in psoriatic skin inflammation, enthesitis, and systemic manifestations such as aortic root and eye involvement [35,69].

Innate lymphoid cells (ILC3s) producing IL-17 are elevated in the peripheral blood and synovial fluid of PsA patients, correlating with disease activity [70,71]. Similarly, MAIT cells produce IL-17, IFN-γ, and TNF and are enriched in psoriatic skin and in the synovial fluid of PsA patients [62]. Invariant natural killer T (iNKT) cells, which recognize lipid antigens via CD1d, also contribute to PsA. Though their role in human PsA is less defined, mouse models demonstrate that iNKT cells drive disease severity and IL-17 production [72].

## 3. Overview of IL-17-Targeting Biologic Therapies in Spondyloarthropathies

IL-17 plays a pivotal role in the pathogenesis of spondyloarthropathies, prompting the development of numerous biologic agents designed to target this pathway.

### 3.1. Secukinumab

Secukinumab targets IL-17A and has demonstrated significant efficacy in managing spondyloarthropathies, including PsA, axSpA, and PsO, and it is accompanied by a favorable safety profile, characterized by a limited incidence of severe adverse events [73,74,75]. It remains an effective option for patients who have failed anti-TNF therapy [76,77]. Clinical studies, such as FUTURE 5, showed that secukinumab achieves a 40–60% resolution in enthesitis and dactylitis, with minimal radiographic progression over three years [78]. In addition, the EXCEED trial revealed superior dermatological outcomes with secukinumab compared to adalimumab, though the rheumatological responses were comparable [79]. In the MEASURE trials, secukinumab’s ability to significantly reduce disease activity in axSpA patients was demonstrated, including improvements in pain, spinal mobility, and inflammation [80]. Sustained benefits have been observed over long-term follow-ups, with minimal radiographic progression reported up to four years [81].

### 3.2. Ixekizumab

Ixekizumab, another IL-17A inhibitor, has demonstrated strong efficacy in treating both axSpA and PsA. In the COAST-V and COAST-W studies, ixekizumab showed significant efficacy in the treatment of both bDMARD-naïve and anti-TNF-experienced patients with axSpA, effectively reducing disease activity, pain, and inflammation [82]. In addition to axSpA, ixekizumab has proven highly effective in managing psoriasis, offering significant improvements in joint symptoms, dactylitis, and skin involvement, providing a holistic benefit for patients suffering from both the dermatologic and musculoskeletal manifestations of the disease. A direct head-to-head comparison in the SPIRIT-H2H study found ixekizumab to be superior to adalimumab in patients with psoriatic arthritis, showing greater improvements in both clinical outcomes and physical function [83].

### 3.3. Brodalumab

Brodalumab is a monoclonal antibody that targets IL-17RA, inhibiting its activation and disrupting downstream inflammatory signaling. In phase 3 trials, brodalumab demonstrated significant efficacy in both axSpA and PsA, though it has not been as extensively studied as other IL-17 inhibitors, such as secukinumab or ixekizumab. In axSpA, a multicenter, placebo-controlled phase 3 study conducted across Japan, Korea, and Taiwan, brodalumab significantly improved disease activity compared to a placebo, with notable enhancements in spinal mobility and overall function. Its efficacy and safety profile were comparable to that of other IL-17 inhibitors previously evaluated in axSpA patients [84]. In PsA, the AMVISION-1 and AMVISION-2 trials demonstrated that brodalumab was significantly more effective in reducing joint pain, swelling, and skin lesions compared to a placebo. Additionally, it led to improvements in dactylitis [85].

### 3.4. Bimekizumab

Bimekizumab is the first humanized IgG1/κ monoclonal antibody that selectively binds to IL-17A, IL-17F, and IL-17AF with high affinity, thereby inhibiting their interaction with the receptor. It was developed based on in vitro findings suggesting that dual inhibition is more effective than targeting IL-17A alone. Consequently, dual neutralization of these IL-17 isoforms may help prevent secondary loss of therapeutic efficacy [86]. Additionally, preclinical studies have highlighted the role of IL-17A and IL-17F in pathological bone formation, showing that dual inhibition of these cytokines can more effectively modulate osteoblastic activity compared to IL-17A blockade alone [87].

Bimekizumab was initially validated in the treatment of psoriasis, where comparative trials, including BE READY (bimekizumab vs. placebo) [88], BE VIVID (bimekizumab vs. ustekinumab) [89], BE RADIANT (bimekizumab vs. secukinumab) [90], and BE SURE (bimekizumab vs. adalimumab) [91], established its superior efficacy compared to anti-TNF agents and other anti-IL-17 therapies. Unfortunately, comparable head-to-head studies of bimekizumab in the field of rheumatology have yet to be conducted.

## 4. Efficacy of Bimekizumab in Spondyloarthropathies

### 4.1. Efficacy of Bimekizumab in Psoriatic Arthritis

The **BE ACTIVE** study, a randomized, placebo-controlled phase IIb trial, evaluates the efficacy of bimekizumab in treating psoriatic arthritis. The primary endpoint was the achievement of an ACR50 response after 12 weeks of treatment with bimekizumab. This study demonstrated a rapid onset of action, with initial improvements observed as early as week 8 and firmly established by week 12. At week 12, 41% of patients receiving bimekizumab at a dose of 160 mg every 4 weeks achieved an ACR50 response, compared to only 7% in the placebo group. Notably, these results are consistent regardless of prior anti-TNF use [92] (Table 1).

The **BE OPTIMAL** study is a 52-week, randomized, double-blind, placebo-controlled phase III trial that includes an active comparator arm with adalimumab. In biologic-naïve patients, bimekizumab demonstrated superior efficacy compared to the placebo across the spectrum of psoriatic arthritis. At week 16, the primary endpoint—an ACR50 response—was achieved in 44% of patients treated with bimekizumab, compared to 10% in the placebo group. The ACR50 response rate in the adalimumab group was similar, at 46%. Moreover, American College of Rheumatology criteria ≥20% (ACR20) and ACR70 responses at week 16 were higher in the bimekizumab group than in the placebo group. These improvements were sustained over a 24-week period, with 45% of patients maintaining an ACR50 response. Among patients with concomitant psoriatic skin disease, 56% achieved complete skin clearance (Psoriasis Area and Severity Index ≥100% (PASI 100)), and 73% achieved a PASI90 by week 24. Clinical responses occurred rapidly, with 27% of patients achieving an ACR20 response within 2 weeks of starting bimekizumab treatment, compared to 7% in the placebo group after a single injection [93].

The **BE COMPLETE** study, a multicenter, double-blind, placebo-controlled, randomized phase III trial, assesses the efficacy of bimekizumab in patients with active psoriatic arthritis who previously experienced inadequate responses or intolerance to anti-TNF therapy. At week 16, 43% of the patients treated with bimekizumab achieved the primary endpoint of an ACR50 response, compared to 7% in the placebo group. Additionally, among patients with baseline psoriasis affecting at least 3% of their body surface area, 69% of those treated with bimekizumab achieved a PASI90 response, compared to 7% in the placebo group [19].

The **BE VITAL** study serves as an extension of the BE COMPLETE and BE OPTIMAL trials, examining the long-term efficacy and safety of bimekizumab over 52 weeks in psoriatic arthritis. At week 52, 59.7% of patients treated with bimekizumab maintained an ACR50 response [94] (Table 1).

### 4.2. Efficacy of Bimekizumab in Axial Spondyloarthritis

The **BE MOBILE 1** study investigates the efficacy of bimekizumab in both r-axSpA and nr-axSpA axial spondyloarthritis. This multicenter, randomized, double-blind, placebo-controlled phase III trial randomized patients at a 1:1 ratio to receive 160 mg of bimekizumab or a placebo every 4 weeks until week 16. The primary endpoint was the achievement of an ASAS40 response (Assessment in SpondyloArthritis international Society 40%), representing at least a 40% improvement in disease activity criteria at week 16. The results showed that ASAS40 responses were achieved in 44.8% of r-axSpA patients and 47.7% of nr-axSpA patients [20,96,97].

The **BE MOBILE 2** study builds on the findings of BE MOBILE 1, extending the evaluation of bimekizumab’s efficacy over a 52-week period. Patients were randomized at a 2:1 ratio to receive 160 mg of bimekizumab or a placebo every 4 weeks until week 16, after which all patients received bimekizumab during a 36-week open-label phase. By week 52, ASAS40 response rates had increased to 50.8% in r-axSpA patients and 68.5% in nr-axSpA patients. Additionally, significant improvements were observed across all components of the ASAS criteria [20,96,97].

The **BE AGILE** study is a randomized, double-blind, placebo-controlled phase IIb trial designed to evaluate the efficacy and safety of bimekizumab in patients with active axSpA. The participants included those with active disease despite prior anti-TNF therapy or intolerance to such agents. The patients were randomized at a 1:1:1:1:1 ratio to receive various doses of bimekizumab (16 mg, 64 mg, 160 mg, or 320 mg) or a placebo for 12 weeks. By week 12, the patients treated with bimekizumab achieved significantly higher ASAS40 response rates compared to the placebo group. Following this, the patients initially receiving 16 mg, 64 mg, or the placebo were re-randomized at 1:1 to receive 160 mg or 320 mg of bimekizumab every 4 weeks until week 48. The response rates were sustained through week 48, with an ASAS40 response rate of 51% [95].

Patients completing the 48-week treatment in the BE AGILE trial were eligible to enroll in the **BE AGILE OLE** (open-label extension) study. The participants received bimekizumab at either 160 mg or 320 mg, administered every 4 weeks, at a 1:1 ratio, until week 156. The results from this extension study demonstrated that bimekizumab’s efficacy was maintained over the long term, with a 57.1% ASAS40 response observed at week 156. No new adverse events were reported throughout the extension period [92] (Table 1).

## 5. Safety Profile

Initial observations suggest a reassuring safety profile across all studied indications. The most frequently reported adverse reactions include upper respiratory tract infections and oral candidiasis, with the incidence varying based on the underlying condition. Upper respiratory tract infections were reported in 14.5% of patients with psoriasis, 14.6% of patients with psoriatic arthritis, and 16.3% of patients with axial spondyloarthritis. Oral candidiasis was observed in 7.3% of patients with psoriasis, 2.3% of patients with psoriatic arthritis, and 3.7% of patients with axSpA. Importantly, no cases of invasive candidiasis have been documented [89,92,98].

Furthermore, no severe or opportunistic infections have been reported in clinical trials. Similarly, no cases of active tuberculosis were observed. It remains crucial, however, to exclude both active and latent tuberculosis before initiating therapy with bimekizumab. In cases of latent tuberculosis, prophylactic anti-tuberculosis treatment should be administered for at least one month prior to starting biologic therapy [89,92,98].

A limited number of neutropenia cases have been documented (1–1.5%) [19,94]. These episodes were mostly transient and did not require discontinuation of therapy. Notably, neutropenia was not associated with severe infections. Mild elevations in liver enzymes have also been reported in isolated cases (0.7–3.7%) [92,94]. In addition, cases of diarrhea and hypertension have also been reported in the BE SURE trial at rates of 3.1–5.1% and 3.8–5.6%, respectively [91].

Research underscores the critical role of IL-17 in maintaining the integrity of the intestinal mucosa, raising theoretical concerns that IL-17 inhibition could increase the risk of inflammatory bowel disease (IBD). However, the occurrence of IBD as an adverse event with anti-IL-17 therapy remains rare. Additionally, evidence suggests that the dual inhibition of IL-17A and IL-17F does not appear to worsen IBD risk compared to IL-17A inhibition alone [99].

Rare cases of severe hypersensitivity reactions, including anaphylactic reactions, have been reported. A small number of neoplasms have been observed in patients treated with bimekizumab, primarily basal cell carcinomas (0–1%) [92,94]. It is worth noting that no dedicated carcinogenicity studies have been conducted to date.

Currently, no data are available regarding the use of bimekizumab during pregnancy or lactation. As a result, its use is not recommended in these circumstances.

## 6. Conclusions

Bimekizumab represents the most recent advancement in IL-17-targeted therapies. The dual inhibition of both IL-17A and IL-17F by bimekizumab may confer a therapeutic advantage over the inhibition of IL-17A alone. Its safety profile aligns closely with that of other anti-IL-17 agents, with the primary risks being upper respiratory tract infections and non-invasive candidiasis, likely attributable to compromised epithelial barrier integrity. Bimekizumab has demonstrated robust efficacy in both psoriatic arthritis and axial spondyloarthritis, encompassing both radiographic and non-radiographic forms, including patients who have previously failed anti-TNF therapy.

Clinical studies have established its non-inferiority compared to adalimumab in terms of efficacy. In the realm of dermatology, trials in psoriasis have demonstrated the superiority of bimekizumab over both adalimumab and secukinumab. However, direct comparative studies between bimekizumab and anti-TNF agents, as well as other anti-IL-17 therapies in rheumatology, have yet to be conducted. Such investigations are essential to ascertain whether these dermatological findings can be extrapolated to spondyloarthropathies.

## Figures and Tables

**Figure 1 ijms-26-02315-f001:**
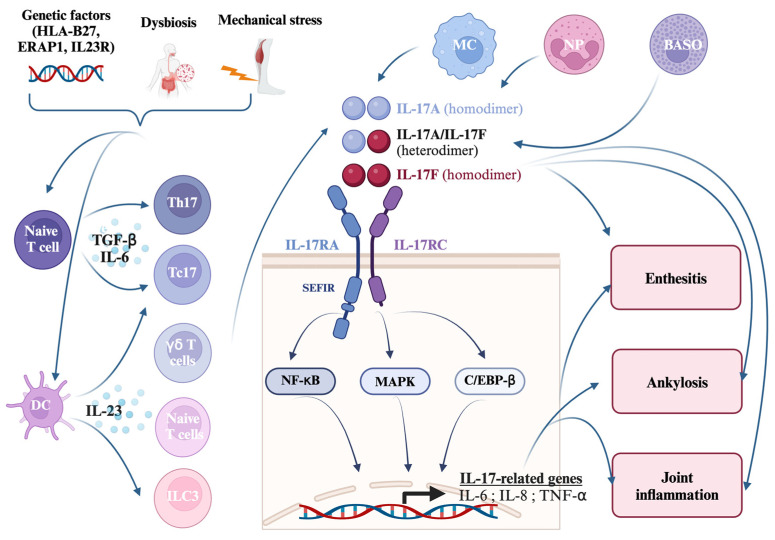
Overview of the IL-17 signaling pathway in the pathophysiology of spondyloarthritis. Genetic predispositions, including HLA-B27, ERAP1, and IL23R, combined with environmental triggers, such as dysbiosis and mechanical stress, lead to the activation of the innate immune system. Under the influence of TGF-β and IL-6, naïve T cells differentiate into Th17 cells, while IL-23 drives the activation of Th17, Tc17, γδ T cells, and ILC3s. Once activated, these cells serve as potent sources of IL-17 cytokines, including IL-17A, IL-17F, and IL-17A/IL-17F heterodimers, which bind to IL-17RA/RC receptors on target cells. Receptor activation triggers intracellular signaling pathways, including NF-κB, MAPK, and C/EBP-β, leading to the expression of pro-inflammatory genes such as IL-6, IL-8, and TNF-α. These cytokines recruit and activate mast cells, neutrophils, and basophils, further amplifying inflammation. This inflammatory cascade drives key pathological features of spondyloarthritis, including enthesitis (inflammation at tendon or ligament attachment sites), ankylosis (abnormal bone formation and joint fusion), and chronic joint inflammation. Abbreviations: BASO: basophils; C/EBP-β CCAAT Enhancer Binding Protein Beta; DCs: dendritic cells; ERAP1: endoplasmic reticulum (ER) aminopeptidase; HLA-B27: human leukocyte antigen B27; IL-: interleukin; IL-23R: IL-23 receptor; IL-17RA: IL-17 receptor A; IL-17RC: IL-17 receptor C; ILC3: group 3 innate lymphoid cells; MAPK: mitogen-activated protein kinase; MC: macrophage; NF-κB: Nuclear factor kappa-light-chain-enhancer of activated B cells; NP: neutrophil; Tc17: IL-17-producing CD8^+^ T cells; Th17: T helper 17 cells; TNF: tumor necrosis factor.

**Table 1 ijms-26-02315-t001:** Summary of key clinical trials evaluating bimekizumab in psoriasis, psoriatic arthritis, and axial spondyloarthritis.

Trial (Phase)	Indication	Design	Primary Endpoints	Results	Ref.
**BE ACTIVE**	PsA	Phase IIb, randomized, placebo-controlled	ACR50 response at 12 weeks	A 41% ACR50 response at 160 mg vs. 7% in the placebo group. Rapid onset of improvement by week 8; consistent results regardless of prior anti-TNF use.	[92]
**BE OPTIMAL**	PsA	Phase III, 52 weeks, randomized, double-blind, placebo or Ada comparator	ACR50 response at 16 weeks	A 44% ACR50 (bimekizumab) response vs. 10% (placebo) vs. 46% (Ada). Sustained responses (45% ACR50 at 24 weeks). PASI 100 achieved in 56% of patients; PASI 90 achieved in 73% of patients. Rapid ACR20 response in 27% of patients by week 2.	[93]
**BE COMPLETE**	PsA	Phase III, randomized, double-blind, placebo-controlled	ACR50 response at 16 weeks	A 43% ACR50 (bimekizumab) response vs. 7% (placebo). Among patients with ≥3% BSA psoriasis, 69% achieved PASI90 vs. 7% in the placebo group.	[19]
**BE VITAL**	PsA	Long-term extension of BE COMPLETE and BE OPTIMAL	ACR50 response at 52 weeks	A 59.7% ACR50 response maintained at 52 weeks.	[94]
**BE MOBILE 1**	nraxSpA	Double-blind, placebo-controlled	ASAS40 response at 16 weeks	A 44.8% ASAS40 (r-axSpA) response and 47.7% ASAS40 (nr-axSpA) response.	[20]
**BE MOBILE 2**	raxSpA	Phase III, 52 weeks, randomized, placebo-controlled	ASAS40 response at 16 and 52 weeks	ASAS40 increased to 50.8% (r-axSpA) and 68.5% (nr-axSpA) by week 52. Significant improvement across all ASAS criteria components.	[20]
**BE AGILE**	axSpA	Phase IIb, randomized, double-blind, placebo-controlled	ASAS40 response at 12 weeks	Higher ASAS40 response rates at 12 weeks for bimekizumab vs. the placebo. Responses sustained through 48 weeks (51% ASAS40).	[95]
**BE AGILE OLE**	axSpA	Open-label extension	ASAS40 response at 48 weeks	Sustained ASAS40 responses, with 52% achieving ASAS40 at 48 weeks. Sustained ASAS responses through week 156.	[92]

Abbreviations: ACR20: American College of Rheumatology criteria ≥ 20%; ACR50: American College of Rheumatology criteria ≥ 50%; Ada: adalimumab; ASAS40: Assessment of SpondyloArthritis international Society ≥ 40%; PASI 90: Psoriasis Area and Severity Index ≥ 90%; nraxSpA: non-radiographic axial spondyloarthritis; PASI 100: Psoriasis Area and Severity Index ≥ 100%; PsA: psoriatic arthritis; rAxSpA: radiographic axial spondyloarthritis; Ref: reference.

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
