# Peer review of "Bimekizumab in the Treatment of Axial Spondyloarthritis and Psoriatic Arthritis: A New Kid on the Block"

_ijms, 2025, doi:10.3390/ijms26052315_

Round 1

Reviewer 1 Report

Comments and Suggestions for Authors

This review article summarized the implications of Bimekizumab in the treatment repertoire of AxSpA and PsA with sufficient details. I have minor comments and please see my comments below. 

  1. duplicate "keywords" section. Please double check.
  2. line 32, remove ")" after AxSpA.
  3. line 40, already mentioned "inflammatory bowel disease (IBD)" above, can use IBD here.
  4. line 88, add the full name of Apso as this is the first time the acronym appears.
  5. line 100 and a global comment, please check AxSpA vs axSpA in the entire article to ensure consistency. 
  6. suggest to mention Figure 1 and Table 1 in the context.
  7. line 311, add "in" before Psoriatic Arthritis?
  8. line 351, please add the full name of "ASAS40".

Author Response

Dear reviewer 

.

thank you for your comments. THey have been implemented in the text 

Kind regards

Reviewer 2 Report

Comments and Suggestions for Authors

In this review, the authors detailed the implications of Bimekizumab (a monoclonal antibody targeting the interleukin IL-17A and IL-17F) in the therapeutic armamentarium of axial spondyloarthritis (axSpA), and psoriatic arthritis (PsA). The authors concluded that Bimekizumab represents the most recent advancement in IL-17-targeted therapies, and suggested that direct comparative studies between bimekizumab and anti-TNF agents, as well as other anti-IL-17 therapies in rheumatology, are necessary to determine whether these dermatological findings can be extrapolated to spondyloarthropathies.

I have some questions and observations:

1.- Figure 1 is described in p 6 but it is not mentioned in the text.

2.- Table 1 is described in p 9  but it is not mentioned in the text

3.-  Does this review follow the Systematic Reviews and Meta-Analyses (PRISMA) statement, published in 2009?. PRISMA was designed to help systematic reviewers transparently report: i) why the review was conducted, ii) what the authors did, iii) what they did find?, and iv) what is the importance of having a review article on this topic?

4.- How many reviews have been published about it?. In this manuscript, there are only 5 and 1 references to works published in 2024 and 2025, respectively. The period covered by the review is not indicated. They carried out a bibliographic review of 98 articles published in the period from 2007 to 2025.

Author Response

Dear reviewer , 

We thank you for your comments and remarks. 

we have implemented your remarks in the text.

This review did not follow the 2009 statement. This review was an invited one and we conducted the study because even if there are already 6 reviews on the subject , they were not as complete and clear . Bimekizumab was reimbursed in mid of 2024 in PSA and SpA. This review will help reading a better grasp of the bimekizumab and its clinical utility .

The method of search was implemented in the text

Kind regards

Reviewer 3 Report

Comments and Suggestions for Authors

This manuscript outlines the key steps involved in Spondyloarthropathies and highlights the common monoclonal antibodies used in recent clinical trials for treatment. The paper is well-written and effectively conveys recent advancements in clinical research. However, numerous recent reviews in this field may diminish the novelty and impact of this review, such

as:

  • https://www.tandfonline.com/doi/abs/10.1080/14740338.2024.2343017
  • https://www.tandfonline.com/doi/abs/10.1080/14740338.2024.2343017
  • https://www.tandfonline.com/doi/full/10.2147/PTT.S367744#d1e259
  • https://link.springer.com/article/10.1007/s13555-022-00760-8
  • https://journals.sagepub.com/doi/abs/10.1177/10600280241288553

The authors should extend the previous published works to make the manuscript attractive in the literature.

Points to consider:

The title can be more general as you described the disease mechanism and included other mAbs.

line 32: consider deleting the right hand side part of the bracket )

line 54-55: include the reference, please.

Line 70: consider deleting “and” at the beginning of the sentence.

Line 365: why these concentrations were tried and what is the optimum to maintain the efficacy?

Table 1: why not to include BE AGILE OLE.

Line 387: hypertension and diarrhea were reported in other studies. You should be thorough when it comes to side effects.

Author Response

Dear Reviewer, 

we thank you for your comments and suggestions .They have been implemented in the text ; 

we have only the title as it is because more than 80% of the text regards bimekizumab. we hope that is ok for you.

kind regards

Round 2

Reviewer 2 Report

Comments and Suggestions for Authors

In this review, the authors detailed the implications of Bimekizumab (a monoclonal antibody targeting the interleukin IL-17A and IL-17F) in the therapeutic armamentarium of axial spondyloarthritis (axSpA), and psoriatic arthritis (PsA). The authors concluded that Bimekizumab represents the most recent advancement in IL-17-targeted therapies, and suggested that direct comparative studies between bimekizumab and anti-TNF agents, as well as other anti-IL-17 therapies in rheumatology, are necessary to determine whether these dermatological findings can be extrapolated to spondyloarthropathies.

I had some questions and observations for the original version of the manuscript:

1.- Figure 1 is described in p 6 but it is not mentioned in the text.

2.- Table 1 is described in p 9  but it is not mentioned in the text

3.-  Does this review follow the Systematic Reviews and Meta-Analyses (PRISMA) statement, published in 2009?. PRISMA was designed to help systematic reviewers transparently report: i) why the review was conducted, ii) what the authors did, iii) what they did find?, and iv) what is the importance of having a review article on this topic?

4.- How many reviews have been published about it?. In this manuscript, there are only 5 and 1 references to works published in 2024 and 2025, respectively. The period covered by the review is not indicated. They carried out a bibliographic review of 98 articles published in the period from 2007 to 2025.

The document has been improved. The new version of the manuscript includes answers to questions and observations.

Reviewer 3 Report

Comments and Suggestions for Authors

Good luck.